# Mip-Grid: Anti-aliased Grid Representations for Neural Radiance Fields

**Seungtae Nam**[1]        **Daniel Rho**[2]        **Jong Hwan Ko**[1,3]
**Eunbyung Park**[1,3*]
[1]Department of Artificial Intelligence, Sungkyunkwan University
[2]AI2XL, KT
[3]Department of Electrical and Computer Engineering, Sungkyunkwan University

## Abstract

Despite the remarkable achievements of neural radiance fields (NeRF) in representing 3D scenes and generating novel view images, the aliasing issue, rendering "jaggies" or "blurry" images at varying camera distances, remains unresolved in most existing approaches. The recently proposed mip-NeRF has addressed this challenge by rendering conical frustums instead of rays. However, it relies on MLP architecture to represent the radiance fields, missing out on the fast training speed offered by the latest grid-based methods. In this work, we present mip-Grid, a novel approach that integrates anti-aliasing techniques into grid-based representations for radiance fields, mitigating the aliasing artifacts while enjoying fast training time. The proposed method generates multi-scale grids by applying simple convolution operations over a shared grid representation and uses the scale-aware coordinate to retrieve features at different scales from the generated multi-scale grids. To test the effectiveness, we integrated the proposed method into the two recent representative grid-based methods, TensoRF and K-Planes. Experimental results demonstrate that mip-Grid greatly improves the rendering performance of both methods and even outperforms mip-NeRF on multi-scale datasets while achieving significantly faster training time. For code and demo videos, please see `https://stnamjef.github.io/mipgrid.github.io/`.

## 1   Introduction

Neural radiance fields (NeRF [19]) have been remarkably successful in modeling and reconstructing 3D scenes. Since its seminal contributions to novel view synthesis, researchers have investigated potential applications of NeRF across various domains in 3D computer vision and graphics, such as dynamic scene representation [43, 24, 25, 15], multi-view image generation [30, 6, 29, 32], 3D surface reconstruction [14], 3D content creation [26, 16, 18, 17], and SLAM [33, 47, 28].

In a typical neural rendering setup, NeRF represents a particular scene, mapping input coordinates $(x, d)$, where $x$ is the spatial coordinate, and $d$ is the view direction, to its corresponding RGB colors and density values $(r, g, b, \sigma)$. Due to its one-to-one mapping, NeRF suffers from aliasing effects when rendering scenes of varying image resolutions; it is incapable of producing different values for different resolutions given the same input coordinates. Mip-NeRF [1] has addressed this issue by introducing integrated positional encoding (IPE), which takes the volume of 3D conical frustums into account. This allows MLPs to reason about the size and shape of each conical frustum, eliminating the inherent ambiguity present in the original NeRF. While this approach provides a successful multi-scale representation, using a variant of positional encoding necessitates MLP architectures for the scene representations, which often leads to a long training time, even with contemporary GPUs.

---

*Corresponding author.

37th Conference on Neural Information Processing Systems (NeurIPS 2023).

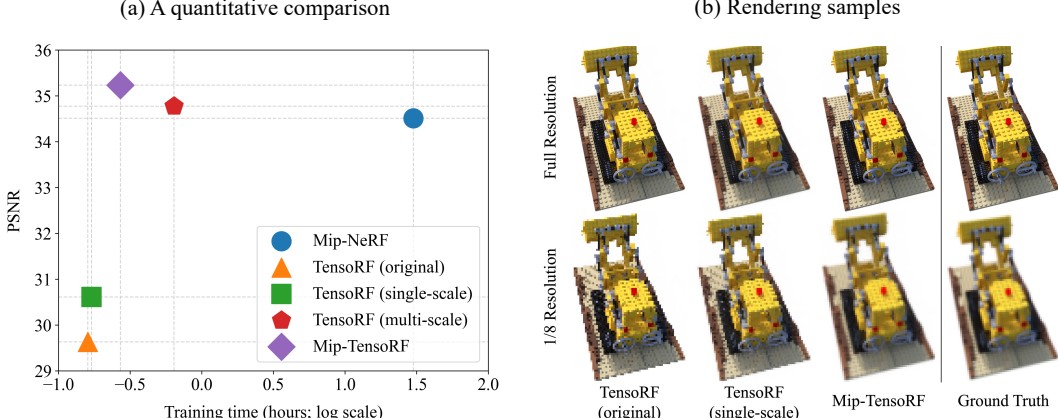

Figure 1: (a) A quantitative comparison of mip-TensoRF against its three baseline models and mip-NeRF (see Sec. 4 for details). (b) Rendering samples of the *lego* scene in two different scales.

To accelerate the training time of NeRF, the grid-based representations with or without MLPs have been extensively studied [9, 21, 8]. These approaches allow faster retrieval of color and density values at given input coordinates, reducing training time to less than an hour while maintaining rendering performance. Furthermore, the recent hashing technique with sophisticated code optimization has demonstrated that the training can be done in a few minutes [21]. However, akin to the original NeRF, these methods still implement the one-to-one mapping between input coordinates and the radiance values, resulting in multi-scale ambiguity. In this paper, we seek to address this limitation, combining grid-based representations for faster training and multi-scale representations for anti-aliasing.

A naive approach to implementing multi-scale representations using grid-based data structures, drawing inspiration from mipmap approaches [41], could involve training separate grid representations for different scales. During the rendering process, we could look up or interpolate the appropriate grid representations. While such an approach could resolve aliasing issues, it would substantially increase the storage requirements. As NeRF has emerged as a new media paradigm, such as 3D photos and videos, large data sizes would unduly waste valuable communication and storage resources.

With only a single-scale grid representation, an effective technique for mitigating aliasing effects would be the multisampling [10] (or supersampling [40]) method. Given the spatial coordinates and scales, we can sample multiple points within the conical frustums and subsequently average features extracted from the single-scale grid representation. This method can reduce the aliasing effects by reasoning scales during the rendering process. However, this approach comes at a high computational cost, directly proportional to the number of samples. Moreover, a similar study with MLP architecture has shown it requires considerable amounts of samples to achieve comparable performance [1].

This paper proposes mip-Grid, a novel method for anti-aliased grid representations that employs a shared single-scale grid representation and a single-sampling approach. We generate multi-scale grid representations by applying convolutions to the shared single-scale grid representation. Then, the features are extracted by interpolating the generated multi-scale grid representations using an additional input coordinate that can reason about the scales. Since the multi-scale grids are generated during the rendering process, the proposed method minimizes additional storage costs. Furthermore, the convolutions to generate multi-scale grids operate on reduced dimensions (2D or 1D instead of 3D convolutions), thereby not incurring substantial computational and memory costs. Lastly, the proposed method can easily be integrated into the recently proposed grid representations, such as factorized tensor representations [7] or plane-based representations [8], with minimal modifications.

We integrated mip-Grid into two state-of-the-art grid-based NeRFs: TensoRF and K-Planes. The resulting models, mip-TensoRF and mip-K-Planes, achieved significantly better performance compared to the original TensoRF and K-Planes, and mip-TensoRF even outperformed mip-NeRF while achieving significantly faster training time (Fig. 1-(a)). We observed that the aliasing artifacts (e.g., "jaggies") are effectively removed from images rendered with the proposed method (Fig. 1-(b)), which are visually more pleasing as well. Additionally, we conducted extensive ablation studies and visualized the learned kernels to help readers understand how the proposed method works internally.

## 2 Related Work

### 2.1 Grid-based neural radiance fields

Representing continuous radiance fields with coordinate-based neural networks, NeRF can synthesize view-consistent and photorealistic novel views. However, it requires a long training time due to the spectral bias inherent in neural network training [27] and the need to query deep neural networks thousands of times to render a single image. A line of work has addressed this issue by encoding input coordinates with Fourier features [37] or learnable positional parameters that are jointly optimized during the training. The latter approach uses various data structures, such as voxel grids [9, 34], octrees [44, 35, 38], and hash tables [21, 39], to store the learnable parameters, each of which is responsible for learning the location-aware features that are decoded into color and density values. Encoding the low-dimensional input coordinates with the learnable parameters significantly accelerates the training and reduces the rendering time, mitigating the inherent spectral bias and eliminating the need to query the deep neural networks.

The dense voxel grid is a straightforward data structure to organize the learnable parameters in 3D space. Plenoxels [9] bake the radiance fields into dense voxel grids, resulting in $100\times$ faster convergence compared to the NeRF. However, cubically increasing memory usage makes the model unscalable. PlenOctrees [44] reduce the memory footprint of Plenoxels at inference time by replacing the dense grids with sparse octrees after the model convergence. This allows the model to discard the empty space of a scene, but the memory requirement still increase cubically during training. TensoRF [7] and other works [8, 5, 4] decompose the dense voxel grids into a set of matrices or vectors, improving parameter efficiency and rendering quality.

The primary focus of our work lies in developing an anti-aliasing framework for decomposed grid representations. While mip-NeRF has addressed the aliasing issue by introducing integrated positional encoding, the discussion was limited to MLP-based radiance fields, and the direct application to grid representations is not straightforward. We show that our method successfully removes the aliasing artifacts in grid representations, significantly reducing the training time compared to mip-NeRF.

### 2.2 Anti-aliasing and mipmap

Representing a discrete signal through sampling requires a higher sampling rate than Nyquist rate [31, 22] (twice the highest frequency of a given signal), and violating this rate occur aliasing artifacts, such as jaggedness and flickering, resulting in a low-quality visualization. Anti-aliasing has been widely studied in computer graphics, and it can be divided into two categories: post-filtering and pre-filtering. Supersampling [40] is one of the most representative post-filtering methods, which renders an image at a higher resolution than the target output resolution. Though effective, supersampling requires high computational and memory costs. The pre-filtering methods [23, 42] reduce the Nyquist rate by filtering a target signal into the low-frequency region before sampling in order to meet the sampling theorem. Mipmap [41] is one of the pre-filtering methods designed for real-time multi-scale rendering. It exploits a pre-filtered hierarchy of progressively lower resolution signals, in which the appropriate level is selected based on the desired scale to render. Mipmap can also represent a scene at arbitrary scales by interpolating the different levels of filtered signals.

Two concurrent works, Zip-NeRF [3] and Tri-MipRF [11], incorporate anti-aliasing techniques into grid-based NeRFs. Zip-NeRF approximates the integration of the grid representations by using multisampling techniques [10], which requires around 50 samplings to query a point during rendering. On the other hand, Tri-MipRF introduces tri-plane mipmaps to represent multi-scale 3D scenes, where the base level mipmap is trained during the reconstruction and the other levels of mipmaps are constructed by downscaling the base level mipmap. Both methods are implemented based on the tiny-cuda-nn [20], a highly optimized deep learning framework, achieving fast training time.

Our work builds on the concept of mipmap in graphics to address the aliasing in grid representations. We share some similarities with Tri-MipRF by incorporating a hierarchy of features that can be decoded into color and density values at queried scales. However, while Tri-MipRF construct the low-level mipmap by downscaling the previous level mipmap by a factor of 2, our method learns the optimal kernels that can generate multi-scale grid representations. Moreover, unlike both Zip-NeRF and Tri-MipRF, we aim to develop a simple framework that can be easily integrated into existing grid-based NeRFs without extensive modifications and hyperparameter tuning.

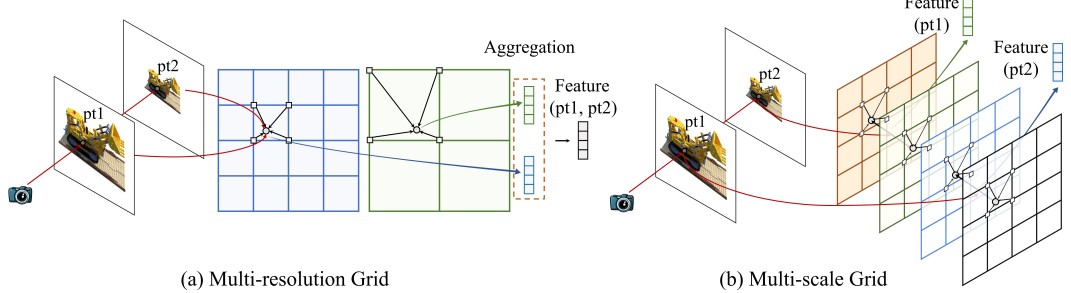

|  | |
|---|---|
| (a) Multi-resolution Grid | (b) Multi-scale Grid |

Figure 2: Multi-resolution and multi-scale grid representations: the pt1 and pt2 are the same spatial coordinates in 3D space. When rendering from varying camera distances, (a) multi-resolution grid representations still retrieve the same features for both pt1 and pt2, while (b) multi-scale grid representations have distinct grids for each pt1 and pt2, hence, effectively resolving scale ambiguity.

## 3 Method

### 3.1 Background: grid representations for neural radiance fields

Grid representations have proven to be highly successful in expediting the training time of NeRF [21, 7, 9]. Combined with lightweight MLPs, typically followed by feature extraction from the grids, it reduced the training times from days to hours or minutes. Furthermore, unlike the original NeRF that relies on MLPs, grid-based approaches do not exhibit spectral bias inherent in neural networks [27]. Through the regularization and training techniques, it often surpasses the performance of the MLPs-based methods while achieving significantly faster training time [21, 7]. In a typical setup, two distinct grids represent density and RGB colors separately. Formally,

$$\sigma = f_\sigma(x; \mathcal{G}_\sigma), \quad c = f_c(x, d; \theta, \mathcal{G}_c), \tag{1}$$

where $\sigma \in [0, \infty)$ is density, $c \in [0, 1]^3$ is RGB colors, $\mathcal{G}_\sigma \in \mathbb{R}^{H \times W \times L \times C_\sigma}$ is a 3D grid for the density features, and $\mathcal{G}_c \in \mathbb{R}^{H \times W \times L \times C_c}$ is a 4D grid for appearance features. $H$, $W$, $L$ are the resolutions of the grids along each spatial axis $X$, $Y$, $Z$, while $C_\sigma$ and $C_c$ are the number of density and appearance feature channels, respectively. Given an input coordinate, $f_\sigma$ interpolates the density features in $\mathcal{G}_\sigma$ and applies a non-linear activation function that outputs density values. Similarly, $f_c$ extracts the appearance features from $\mathcal{G}_c$, and normally takes an additional directional input $d$, and then these inputs are passed through an MLP to generate RGB colors. The final RGB colors for a given ray are generated using differentiable volume rendering [12], and simple reconstruction loss is used to train the $\mathcal{G}_\sigma$, $\mathcal{G}_c$, and the parameters of the MLP $\theta$.

### 3.2 Multi-resolution and multi-scale grid representations

Many recent studies have successfully utilized multi-resolution (or multi-level) grid representations to encode various signals [21, 8]. To avoid confusion, we would like to clarify that multi-resolution grids refer to the usage of grids with different resolutions, while multi-scale grids refer to grids tailored to each image scale. As illustrated in Fig. 2-(a), the multi-resolution grid representations are designed to extract features from multiple grids with different resolutions, followed by aggregating operations to combine the extracted features, such as concatenation, addition, or multiplication. This approach draws inspiration from signal processing, encouraging the low-resolution grids to capture global and smooth parts of the signal (low-frequency components), while high-resolution grids represent fine details of the signal (high-frequency components). This decomposed representation is often parameter efficient and better reconstructs the signals. In addition, the recent hashing-based technique [21] also leverages multi-resolution grids to disambiguate hash collisions.

However, due to its inherent one-to-one mapping nature, the current multi-resolution grid representations have limitations in addressing the aliasing issue. When provided with identical input coordinates, the same features are extracted regardless of the camera distances (or scales). Consequently, the grid representations tend to learn to render the average scale images when training on multi-scale images (or images from varying camera distances). This results in "blurred" images when rendering higher resolutions than the average training resolutions and "jaggies" when rendering at lower resolutions.

Fig. 2-(b) depicts multi-scale grid representations, where individual grids (or groups of neighboring grids) are responsible for different scale images. In this setup, features are exclusively extracted from a single grid or interpolated from neighboring grid representations. As a result, this method can effectively address scale ambiguity by introducing distinct grids for different scales. While a promising approach, it dramatically increases the space complexity, making it an impractical solution for many potential NeRF applications, excessively exhausting communication and storage resources.

### 3.3 Shared grid and generating multi-scale grids

Since having distinct grid representations for different scales is not a viable solution, we propose to generate multi-scale grids with minimal storage requirements. Inspired by mipmapping techniques commonly used in computer graphics [41], and also motivated by the fact that a significant amount of information is shared across different scales, we use a shared grid representation and generate multi-scale grids by applying a simple convolution over the shared grid. The modified radiance fields with the generated multi-scale grids can be written as follows:

$$\sigma = f_\sigma(x, s; \tilde{\mathcal{G}}_\sigma), \quad c = f_c(x, s, d; \theta, \tilde{\mathcal{G}}_c), \tag{2}$$

where $\tilde{\mathcal{G}}_\sigma \in \mathbb{R}^{H \times W \times L \times S \times C_\sigma}$ is the generated multi-scale grids for the density features, and $\tilde{\mathcal{G}}_c \in \mathbb{R}^{H \times W \times L \times S \times C_c}$ is the generated multi-scale grids for the appearance features. $S$ is the number of the generated multi-scale grids, and $s$ is a scale-aware input coordinate.

There are several options for generating $\tilde{\mathcal{G}}$. We can simply use Gaussian filtering, which has been widely used in conventional computer graphics. However, it may not be the optimal filter for many real-world scenarios. Alternatively, we can utilize convolution with learnable filters or leverage multi-layer neural networks such as convolutional neural networks (CNNs) or Transformers. Although complex neural networks can potentially yield better multi-scale representations, there is a trade-off in terms of computational complexity and performance. This work primarily investigates a single convolutional layer within our proposed multi-scale representations.

### 3.4 Scale-aware input coordinates

In order to reason about the scale, mip-NeRF introduced the integrated positional encoding to augment input coordinates to disambiguate the radiance fields from different camera distances: the original NeRF: $(x, d) \rightarrow (\sigma, c)$ and mip-NeRF: $(x, d, \sigma_t^2, \sigma_r^2) \rightarrow (\sigma, c)$, where $\sigma_t^2 \in \mathbb{R}$ and $\sigma_r^2 \in \mathbb{R}$ are variances of the conical frustum with respect to its points interval along the ray and radius, respectively. As a result, mip-NeRF can provide different radiance values for the same coordinates $(x, d)$, reflecting the radii of sampled points $(\sigma_t^2, \sigma_r^2)$ to effectively mitigate the aliasing artifacts.

In a similar vein, we introduce additional scale-aware coordinate, which is used to extract the features from the generated multi-scale grid representations. Specifically, we introduce three types of scale coordinates: the discrete scale coordinate, the continuous scale coordinate, and the 2-dimensional scale coordinate. The discrete scale coordinate is defined as the mean of the pixel width and height in the world coordinates scaled by $2/\sqrt{12}$ [1]. Since this value is unique for each image scale, we can simply use the same value for all input coordinates (i.e., sampled points) of each image scale to extract the features, not introducing extra computation to get per-point scale coordinates.

The continuous scale coordinate is the discrete scale coordinate scaled by the distance from the ray origin to the sampled point. Training the model with the discrete scale coordinate alone can lead to overfitting to given scales, making it challenging for the model to render images at unseen scales. For instance, if a dataset includes four scales of images, the model might generate multi-scale grids that overfit to those specific scales. In contrast, the continuous scale coordinate is not unique for each image scale. Each sampled point has a distinct scale value, which allows the model to generate multi-scale grids that can be smoothly interpolated into features across a broader range of scales.

The 2-dimensional scale coordinate consists of the continuous scale coordinate and the distance from the ray origin to the sampled point. While reflecting the discrete or continuous scale coordinate can effectively resolves scale ambiguity, we introduce the third one to provide readers with an understanding of how high-dimensional scale coordinates can be integrated into our method. During the rendering process, we apply convolution to the shared grid using two sets of kernels. We then independently extract features from each multi-scale grids based on the respective scale coordinates and average the feature sets to get the final features.

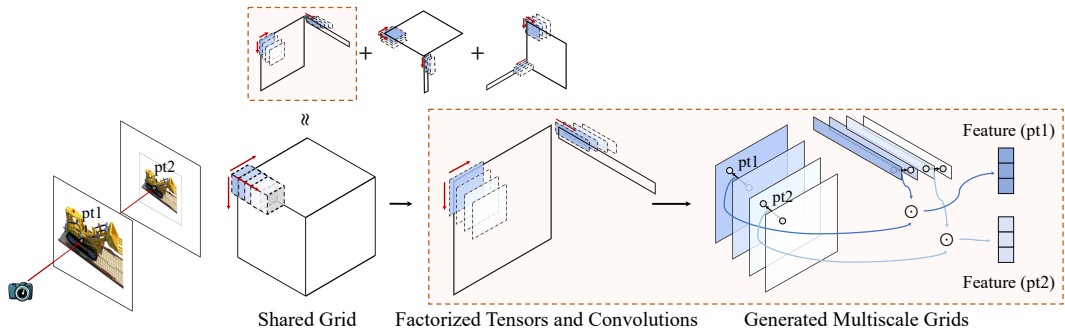

Figure 3: The overall feature extraction pipeline in the proposed mip-TensoRF.

### 3.5 Mip-Grid

Current grid-based NeRFs typically involves certain versions of tensor factorization schemes. For example, TensoRF used CP decomposition for compact models and proposed a new VM (vector-matrix) factorization method for better rendering performance. K-Planes used plane-based factorization, which has been very successful not only in 3D, but also in 2D and 4D scene generation. Thus, we have integrated the proposed method into two prominent grid-based NeRFs: TensoRF and K-Planes.

#### 3.5.1 Mip-TensoRF

In TensoRF, VM factorization for a density grid $\mathcal{G}$ is formulated as follows (we omitted the subscript $\sigma$ for brevity):

$$\mathcal{G} = \sum_{r=1}^{R} v_r^X \circ M_r^{YZ} + v_r^Y \circ M_r^{XZ} + v_r^Z \circ M_r^{XY}, \tag{3}$$

where $v_r^X \in \mathbb{R}^H$ denote the vector for $X$ axis, $M_r^{YZ} \in \mathbb{R}^{W \times L}$ is the matrix for $Y$ and $Z$ axes, $R$ is the rank, and $\circ$ is a tensor product, hence $v_r^X \circ M_r^{YZ} \in \mathbb{R}^{H \times W \times L}$. We generate the multi-scale vectors and matrices by applying 1D or 2D depth-wise convolutions over vectors and matrices. This is equivalent to applying 3D convolutions over 3D tensors thanks to the linearity of convolution operators. The $i$-th scale of a multi-scale grid $\tilde{\mathcal{G}} \in \mathbb{R}^{H \times W \times L \times S}$ can be defined as follows:

$$\tilde{\mathcal{G}}_i = \sum_{r=1}^{R} (v_r^X * k_{i,r}^{v^X}) \circ (M_r^{YZ} * k_{i,r}^{M^{YZ}}) + (v_r^Y * k_{i,r}^{v^Y}) \circ (M_r^{XZ} * k_{i,r}^{M^{XZ}}) + (v_r^Z * k_{i,r}^{v^Z}) \circ (M_r^{XY} * k_{i,r}^{M^{XY}}), \tag{4}$$

where $*$ is the convolution operator, $\tilde{\mathcal{G}}_i \in \mathbb{R}^{H \times W \times L}$ is $i$-th scale of the grid, $k_i^{v^X} \in \mathbb{R}^{K \times R}$ is $i$-th depth-wise convolution filter for the vector $v_r^X$, $k_{i,r}^{v^X} \in \mathbb{R}^K$ is the filter for the rank $r$, and $K$ is the kernel size. Similarly, $k_i^{M^{YZ}} \in \mathbb{R}^{K \times K \times R}$ is $i$-th depth-wise convolution filter for the matrix $M_r^{YZ}$ and $k_{i,r}^{M^{YZ}} \in \mathbb{R}^{K \times K}$ is the filter for the rank $r$. Note that we have separate learnable kernels for each rank, axis, and scale, which are more expressible than fixed Gaussian filters.

#### 3.5.2 Mip-K-Planes

Similar to mip-TensoRF, we apply depth-wise convolutions over the plane-based grid representations. In K-Planes, a grid $\mathcal{G}^k \in \mathbb{R}^{D_x \times D_y \times D_z \times R}$ can be formulated as follows:

$$\mathcal{G}^k = M^{YZ} \odot M^{XZ} \odot M^{XY}, \tag{5}$$

where $M^{YZ} \in \mathbb{R}^{1 \times D_y \times D_z \times R}$, $M^{XZ} \in \mathbb{R}^{D_x \times 1 \times D_z \times R}$, and $M^{XY} \in \mathbb{R}^{D_x \times D_y \times 1 \times R}$ denote three grids for 3D scene representations. $\odot$ denotes broadcast element-wise multiplication. For example, $M^{YZ} \odot M^{XZ}$ will have the shape of $\mathbb{R}^{D_x \times D_y \times D_z \times R}$. $D_x$, $D_y$, and $D_z$ denote the grid resolutions in X, Y, and Z axes. Similar to Eq. 4, a multi-scale grid $\tilde{\mathcal{G}}_i^k \in \mathbb{R}^{D_x \times D_y \times D_z \times R}$ for $i$-th scale can be written as follows:

$$\tilde{\mathcal{G}}_i^k = (M^{YZ} * k_i^{YZ}) \odot (M^{XZ} * k_i^{XZ}) \odot (M^{XY} * k_i^{XY}), \tag{6}$$

where $k_i^{YZ} \in \mathbb{R}^{1 \times K \times K \times R}$, $k_i^{XZ} \in \mathbb{R}^{K \times 1 \times K \times R}$, and $k_i^{XY} \in \mathbb{R}^{K \times K \times 1 \times R}$ denote $i$-th depth-wise convolution filters.

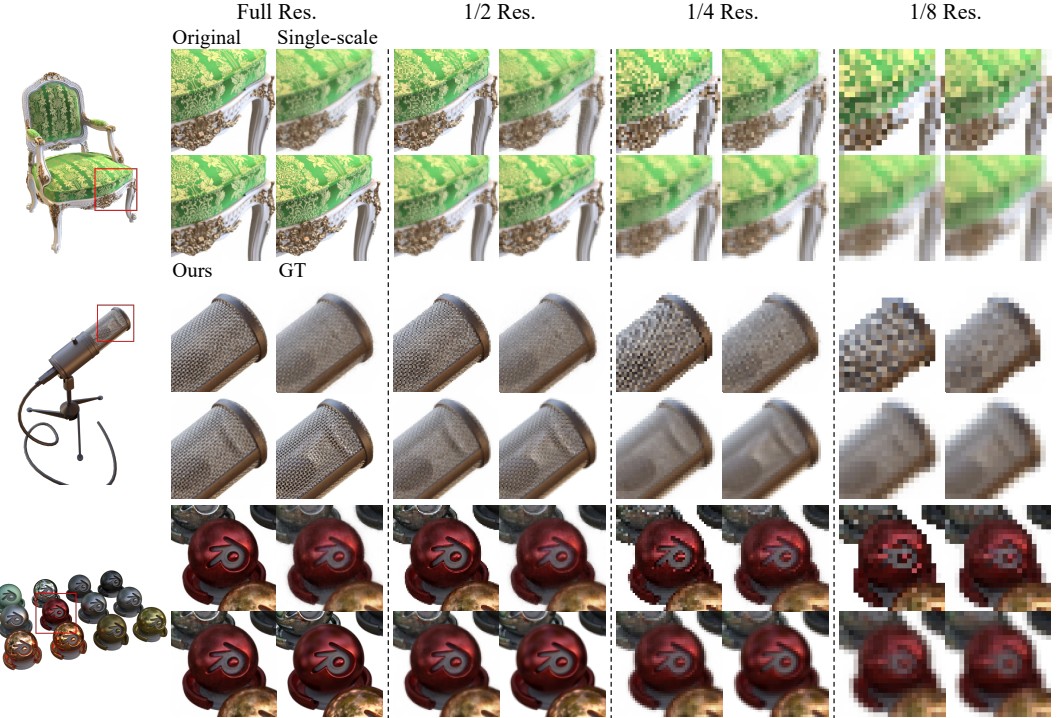

Figure 4: A qualitative comparison of mip-TensoRF against the original TensoRF and the single-scale TensoRF. The zoom-in parts of rendering samples in four different scales are shown. As resolution decreases, two baseline models exhibit aliasing artifacts. Best viewed in color and zoom-in.

## 4 Experiment

### 4.1 Experimental setup

**Implementation details** We implemented two variants of mip-Grid: mip-TensoRF and mip-K-Planes. Both models apply convolution to a shared grid representation using four learnable kernels of size 3, where each kernel is responsible for generating different scales of grids. We trained mip-TensoRF for 40k iterations on the multi-scale Blender dataset and 25k iterations on the multi-scale LLFF dataset. All of its three baseline models were also trained for the same number of iterations. We scaled the loss of each pixel following mip-NeRF [1] to account for the scale differences. Note that we began kernel training after the grid upsampling [7] was completed. Unlike mip-TensoRF, we trained mip-K-Planes for 30k iterations and optimized the kernels from the beginning. All other training hyperparameters remained the same as the original TensoRF-VM-192 and K-Planes.

As outlined in the Sec. 3, mip-Grid introduces three kinds of scale coordinates to extract features from the generated multi-scale grids. We used the discrete scale coordinate to train mip-TensoRF and mip-K-Planes, and for mip-TensoRF, we conducted a further evaluation using the continuous scale coordinate and the 2D scale coordinate, examining the effectiveness of using more sophisticated scale coordinates. In all tables showing the results of mip-Grid integrated models, abbreviations of the scale coordinate types are provided in parentheses along with the name of each model.

**Baseline models** We compare our method against three baseline models: the original model, the single-scale model, and the multi-scale model. The original models are the same as the TensoRF-VM-192 and K-Planes, but we test them on the multi-scale dataset. The single-scale model is trained on the multi-scale dataset with loss multipliers [1] and tested on the multi-scale dataset. The multi-scale model is a set of four independent original models, each of which is trained and tested on different scales of dataset. We include the latter two baseline models to show that naively training the single-scale representations with multi-scale dataset cannot remove aliasing artifacts effectively, and using four independent models requires four times more parameters than the original model.

Table 1: A model evaluation on the test set of multi-scale Blender dataset. The elapsed training time, evaluated using a single RTX 4090 GPU, are shown at the rightmost column. For mip-NeRF, the time elapsed for 100 iterations were measured and multiplied by 10000 to estimate the total runtime for 1 million iterations. Note that the multi-scale model has four times more parameters than the original model.

| | Avg. | PSNR↑ | | | | SSIM↑ | | | | LPIPS↓ | | | | Time (hours) |
|---|---|---|---|---|---|---|---|---|---|---|---|---|---|---|
| | | Full Res. | ½ Res. | ¼ Res. | ⅛ Res. | Full Res. | ½ Res. | ¼ Res. | ⅛ Res. | Full Res. | ½ Res. | ¼ Res. | ⅛ Res. | |
| Mip-NeRF [1] | 34.51 | 32.63 | 34.34 | 35.47 | 35.06 | 0.9579 | 0.9703 | 0.9786 | 0.9833 | 0.0469 | **0.0260** | **0.0168** | **0.0120** | >30 |
| TensoRF (original) | 30.70 | 33.13 | 33.27 | 30.02 | 26.40 | 0.9630 | 0.9704 | 0.9624 | 0.9358 | 0.0474 | 0.0337 | 0.0488 | 0.0802 | 0.16 |
| TensoRF (single-scale) | 30.61 | 28.98 | 32.42 | 32.53 | 28.52 | 0.9295 | 0.9618 | 0.9703 | 0.9514 | 0.1054 | 0.0608 | 0.0474 | 0.0697 | 0.17 |
| TensoRF (multi-scale) | 34.77 | **33.33** | **35.74** | 35.44 | 34.57 | **0.9637** | 0.9757 | 0.9766 | 0.9775 | **0.0456** | 0.0301 | 0.0279 | 0.0238 | 0.66 |
| Mip-TensoRF | **35.23** | 32.53 | 35.40 | **36.58** | **36.41** | 0.9587 | **0.9757** | **0.9828** | **0.9857** | 0.0533 | 0.0278 | 0.0200 | 0.0163 | 0.27 |
| K-Planes (original) | 29.76 | 32.40 | 31.83 | 28.95 | 25.85 | 0.9615 | 0.9663 | 0.9570 | 0.9321 | 0.0491 | 0.0369 | 0.0539 | 0.0861 | 0.51 |
| K-Planes (single-scale) | 30.07 | 32.13 | 32.35 | 29.54 | 26.24 | 0.9590 | 0.9667 | 0.9597 | 0.9356 | 0.0525 | 0.0361 | 0.0497 | 0.0813 | 0.51 |
| K-Planes (multi-scale) | 32.24 | 32.59 | 33.60 | 32.34 | 30.58 | 0.9605 | 0.9696 | 0.9646 | 0.9580 | 0.0508 | 0.0314 | 0.0396 | 0.0415 | 2.02 |
| Mip-K-Planes | 32.27 | 33.20 | 33.03 | 32.73 | 31.12 | 0.9601 | 0.9691 | 0.9724 | 0.9689 | 0.0512 | 0.0329 | 0.0291 | 0.0302 | 0.60 |

Table 2: A model evaluation on the test set of multi-scale LLFF dataset. Note that the multi-scale model has four times more parameters than the original model.

| | Avg. | PSNR↑ | | | | SSIM↑ | | | | LPIPS↓ | | | |
|---|---|---|---|---|---|---|---|---|---|---|---|---|---|
| | | Full Res. | ½ Res. | ¼ Res. | ⅛ Res. | Full Res. | ½ Res. | ¼ Res. | ⅛ Res. | Full Res. | ½ Res. | ¼ Res. | ⅛ Res. |
| TensoRF (original) | 26.03 | 26.73 | 27.89 | 26.70 | 22.81 | **0.8386** | 0.8932 | 0.8964 | 0.8063 | 0.2044 | 0.1069 | 0.1099 | 0.1685 |
| TensoRF (single-scale) | 27.33 | 24.48 | 28.25 | 29.92 | 26.64 | 0.7457 | 0.8863 | 0.9375 | 0.8964 | 0.3114 | 0.1450 | 0.0851 | 0.1259 |
| TensoRF (multi-scale) | 30.18 | **26.75** | 29.58 | 31.33 | 33.06 | 0.8378 | 0.9067 | 0.9487 | 0.9731 | **0.1992** | 0.1065 | 0.0632 | 0.0339 |
| Mip-TensoRF | **30.48** | 26.51 | **29.69** | **32.18** | **33.52** | 0.8310 | **0.9165** | **0.9590** | **0.9762** | 0.2152 | **0.0946** | **0.0477** | **0.0316** |

**Datasets**   All models were evaluated on the multi-scale Blender dataset [1] with three different metrics: PSNR, SSIM, and LPIPS (VGG) [46]. Mip-TensoRF and its baseline models were further evaluated on the multi-scale LLFF dataset. To convert single-scale dataset to multi-scale dataset, we bilinearly downsampled the images by a factor of 1, 2, 4, and 8, and rescaled the focal length accordingly as typically done in other multi-scale NeRF literature [1, 3].

## 4.2   Results

Tab. 1 shows the evaluation results on the multi-scale Blender dataset. Mip-TensoRF and mip-K-Planes outperform their baseline models, achieving the highest average PSNR. Notably, mip-TensoRF even surpasses mip-NeRF while significantly reducing the training time. The performance of the original models experiences a substantial drop as the rendering resolution decreases, leading to noticeable "jaggies" in the images rendered at the two lowest resolutions (Fig. 4). The single-scale models achieve better performance at the two lowest resolutions compared to the original models but show worse results at other resolutions. This suggests that, to some extent, training with a multi-scale dataset and employing the loss multipliers helps the model in reasoning about different scales, but this approach does not entirely eliminate aliasing artifacts. On the other hand, the multi-scale models can effectively represent different scales of a scene, achieving the best PSNR among the three baseline models. Nevertheless, these models require four times more parameters than the original models, which is undesirable in practical applications. Unlike others, our models successfully mitigate aliasing artifacts while minimizing the additional number of parameters and computational cost.

We provide additional results of training mip-TensoRF with continuous and 2-dimensional scale coordinates in Tab. 3. The first two models show only a marginal difference when evaluated at the training resolutions. A similar trend is observed in evaluations at unseen resolutions, but there is a clear difference when the image resolution is reduced from 4/8 resolution to 3/8 resolution, suggesting that the continuous scale coordinate improves the model's generalization performance. Our method generates multiple sets of feature grids from a shared representation, and these feature sets share many similarities. This allows different features to be smoothly interpolated and rendered into visually pleasing images. However, more information needs to be removed as the output images decrease in size, leading to relatively large differences between low-scale features. This explains the marginal drop in PSNR at lower resolutions. Moreover, the model trained with 2D scale coordinate outperforms the others, indicating that the additional scale coordinate (the distance from the ray origin to the sampled point) can be beneficial for resolving the remaining scale ambiguity.

Table 3: An evaluation of mip-TensoRF trained with three different types of scale coordinates. All models were trained on the multi-scale Blender dataset (with four different resolutions; Full Res., 4/8 Res., 2/8 Res., and 1/8 Res.) and tested on the test set of Blender dataset rescaled to eight resolutions.

| | Avg. | Full Res. | 7/8 Res. | 6/8 Res. | 5/8 Res. | 4/8 Res. | 3/8 Res. | 2/8 Res. | 1/8 Res. |
|---|---|---|---|---|---|---|---|---|---|
| | | | | | PSNR↑ | | | | |
| Mip-TensoRF (disc.) | 34.54 | 32.53 | 33.73 | 34.20 | 34.70 | 35.40 | 34.33 | 35.93 | 35.50 |
| Mip-TensoRF (cont.) | 34.73 | 32.46 | 33.82 | 34.29 | 34.74 | 35.34 | 35.64 | 35.89 | 35.66 |
| Mip-TensoRF (2D) | **34.88** | **32.56** | **33.94** | **34.43** | **34.90** | **35.53** | **35.77** | **36.09** | **35.80** |

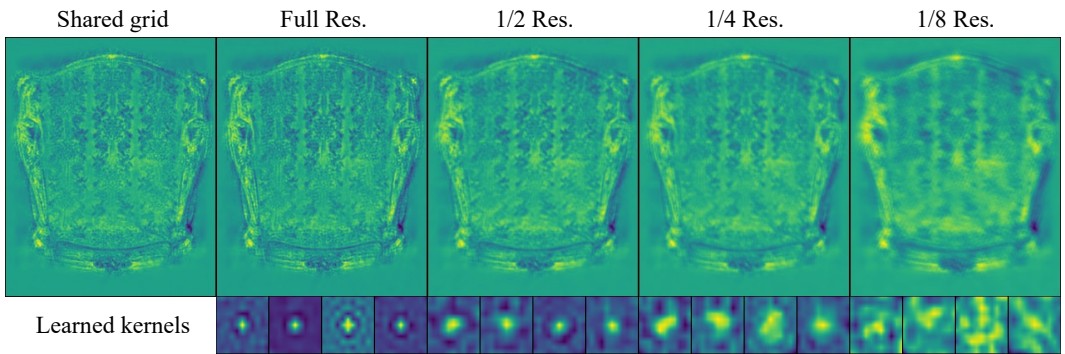

Figure 5: Visualization of generated multi-scale grids and learned kernels of mip-TensoRF. A single channel in the $XY$ plane of the appearance grids and the first four channels of the learned kernels are shown. The first column of the first row displays a shared grid representation, and the generated multi-scale grids are displayed from the second to the last column of the first row. Each column of the second row shows the learned kernels that is used to generate the corresponding feature grids.

## 4.3 Visualization of learned kernels and multi-scale grids

We visualize the learned kernels and the generated multi-scale grids of mip-TensoRF to illustrate how the learned patterns vary across scales. As depicted in Fig. 5, the high-scale kernels look similar to Gaussian distribution, and their shape (i.e., the standard deviation) becomes larger as the corresponding scale decreases. This observation aligns with our intuition that the kernels of low-scale features should be smoother than those of high-scale features and demonstrates that our method effectively learns optimal kernels for each scale. A similar phenomenon can be observed in the visualization of feature grids, where high-frequency details are progressively removed as the feature scale decreases. Although we do not provide the visualization of density kernels and grids due to the limited space, the learned patterns closely resemble those of the appearance kernels and grids.

## 4.4 Ablation studies

**Fixed Gaussian kernels and learnable kernels** We compare mip-TensoRF with fixed Gaussian kernels against the same model with learnable kernels. Both models were initialized with 2 sets of standard deviations {1.0, 1.0, 1.0, 1.0} and {1.0, 1.5, 2.5, 4.0}, where the leftmost value was used to set the kernel of the highest scale, and the rightmost value was used for the lowest scale. The second set is to give an inductive bias that the low-scale features should be smoothed more than high-scale features. The fixed Gaussian kernel initialized with the second set of standard deviations significantly outperforms the other (Tab. 4). This suggests that increasing standard deviations can be a beneficial inductive bias. However, these fixed Gaussian kernels may not be optimal for each scale. Instead of using heuristically selected kernels, our method fine-tunes the Gaussian kernels and finds the optimal weights during the training. Experimental results demonstrate that the learnable approach outperforms the fixed Gaussian approach, further confirming the effectiveness of our method (Tab. 4).

**The number of multi-scale grids and kernel size** The proposed method involves two key hyperparameters: the number of multi-scale grids and the kernel size. We provide an evaluation of mip-TensoRF with nine different combinations of these two values to give readers guidance on hyperparameter configuration. As shown in Tab. 5, the number of multi-scale grids has little effect

Table 4: An evaluation of mip-TensoRF using fixed Gaussian kernels and learnable kernels, initialized with different standard deviations. Trained and tested on the multi-scale Blender dataset.

| | Stdev. | Avg. | PSNR↑ | | | | SSIM↑ | | | | LPIPS↓ | | | |
|---|---|---|---|---|---|---|---|---|---|---|---|---|---|---|
| | | | Full Res. | ½ Res. | ¼ Res. | ⅛ Res. | Full Res. | ½ Res. | ¼ Res. | ⅛ Res. | Full Res. | ½ Res. | ¼ Res. | ⅛ Res. |
| Fixed | 1.0; 1.0; 1.0; 1.0 | 30.99 | 28.59 | 32.46 | 33.75 | 29.15 | 0.9267 | 0.9627 | 0.9763 | 0.9574 | 0.0991 | 0.0494 | 0.0332 | 0.0593 |
| | 1.0; 1.5; 2.5; 4.0 | 34.05 | 30.84 | 34.10 | 35.37 | 35.87 | 0.9478 | 0.9698 | 0.9787 | 0.9829 | 0.0697 | 0.0355 | 0.0255 | 0.0211 |
| Learnable | 1.0; 1.0; 1.0; 1.0 | 35.31 | 32.57 | 35.43 | 36.64 | 36.59 | 0.9588 | 0.9759 | 0.9829 | 0.9863 | 0.0535 | 0.0275 | 0.0196 | 0.0150 |
| | 1.0; 1.5; 2.5; 4.0 | **35.40** | **32.62** | **35.47** | **36.70** | **36.81** | **0.9593** | **0.9762** | **0.9832** | **0.9868** | **0.0529** | **0.0272** | **0.0192** | **0.0141** |

Table 5: An evaluation of mip-TensoRF varying two hyperparameters: N - the number of multi-scale grids, K - the kernel size. Tested on the *hotdog* scene of the multi-scale Blender dataset.

| N | K | Avg. | PSNR↑ | | | | SSIM↑ | | | | LPIPS↓ | | | |
|---|---|---|---|---|---|---|---|---|---|---|---|---|---|---|
| | | | Full Res. | ½ Res. | ¼ Res. | ⅛ Res. | Full Res. | ½ Res. | ¼ Res. | ⅛ Res. | Full Res. | ½ Res. | ¼ Res. | ⅛ Res. |
| 2 | 3 | 39.43 | 36.84 | 39.72 | 40.66 | 40.49 | 0.9792 | 0.9887 | 0.9917 | 0.9931 | 0.0379 | 0.0164 | 0.0104 | 0.0078 |
| | 5 | 39.58 | 36.84 | 39.78 | 40.86 | 40.85 | 0.9790 | 0.9888 | 0.9918 | 0.9934 | 0.0382 | 0.0163 | 0.0099 | 0.0069 |
| | 11 | 39.63 | 36.83 | 39.81 | 40.89 | 41.01 | 0.9789 | 0.9890 | 0.9920 | 0.9937 | 0.0381 | 0.0160 | 0.0096 | **0.0066** |
| 3 | 3 | 39.58 | 37.01 | 39.84 | 40.89 | 40.57 | 0.9795 | 0.9888 | 0.9918 | 0.9932 | 0.0375 | 0.0164 | 0.0102 | 0.0080 |
| | 5 | 39.66 | 36.98 | 39.90 | 40.98 | 40.78 | 0.9796 | **0.9890** | 0.9920 | 0.9935 | 0.0367 | **0.0155** | 0.0095 | 0.0069 |
| | 11 | 39.65 | 36.92 | 39.83 | 40.95 | 40.90 | 0.9793 | 0.9890 | 0.9920 | 0.9937 | 0.0371 | 0.0157 | **0.0093** | 0.0067 |
| 4 | 3 | 39.60 | 37.03 | 39.90 | 40.92 | 40.57 | 0.9797 | 0.9888 | 0.9919 | 0.9932 | 0.0367 | 0.0161 | 0.0099 | 0.0079 |
| | 5 | 39.74 | **37.06** | **39.97** | 41.04 | 40.89 | **0.9798** | 0.9890 | 0.9920 | 0.9936 | 0.0365 | 0.0158 | 0.0095 | 0.0069 |
| | 11 | **39.78** | 37.02 | 39.97 | **41.07** | **41.06** | 0.9796 | 0.9890 | **0.9921** | **0.9938** | **0.0363** | 0.0158 | 0.0093 | 0.0068 |

on the rendering quality, but variations in kernel sizes result in relatively bigger differences in the PSNR of low-scale images. Although different scales of grid representations share many similarities, the low-scale grid should be smoothed enough to produce an image without aliasing artifacts, and the kernel size is a critical factor controlling the smoothness of the grids. Using too small kernels may introduce unnecessary high-frequency components in the low-scale grid. Nevertheless, there is always a trade-off between rendering quality and speed, and users should consider their specific usage purpose when determining the kernel size.

## 5  Limitations and discussion

The proposed method has not been tested on scenes involving more complex camera poses, such as 360-degree scenes [2, 45, 36] or free trajectories [39]. However, addressing these scenarios would require further technical development and a thorough parameter tuning. We plan to investigate how the proposed method, or even the grid representation itself, can be extended to these scenarios.

Although the proposed method is significantly faster than mip-NeRF, it introduces additional complexity to the system. Especially in terms of training time, our method needs more computational resources to generate multi-scale grids and retrieve features. We implemented our model based on TensoRF and K-Planes with minimal modifications, which makes the training time heavily rely on the base models. Also, the implementation of the multi-scale grid generation and the feature retrieval process has not been optimized yet. We strongly believe the training time and memory requirements can be substantially improved by fusing the multi-scale grid generation and the feature retrieval process as a single CUDA kernel or using the NeRF accelerating tool [13].

## 6  Conclusion

We proposed mip-Grid, anti-aliased grid representations for NeRF. The proposed approach can easily be integrated into the existing grid-based NeRFs, and the two methods using our approach, mip-TensoRF and mip-K-Planes, have demonstrated that aliasing artifacts can be effectively removed. Since we generate multi-scale grids from a shared grid representation and do not rely on supersampling, the proposed method minimizes additional number of parameters and trains significantly faster than the existing MLP-based anti-aliased NeRFs. We believe our work paves the way for a new research direction toward alias-free NeRF leveraging the training efficiency of grid representations.

## Acknowledgments and Disclosure of Funding

This work was supported by the Institute of Information and Communication Technology Planning Evaluation (IITP) grants (IITP-2019-0-00421, IITP-2021-0-02068) and the National Research Foundation (NRF) grants (2022R1F1A1064184, RS-2023-00245342) funded by the Ministry of Science and ICT (MSIT) of Korea.

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
