# OpenReview forum: "Mip-Grid: Anti-aliased Grid Representations for Neural Radiance Fields"
_NeurIPS.cc/2023/Conference — NeurIPS 2023 poster_

### Official Review · Reviewer_QWBQ · 2023-07-03

**Soundness:** 3 good
**Presentation:** 3 good
**Contribution:** 2 fair
**Rating:** 6
**Confidence:** 4

**Summary:**

This paper studies the anti-alias problem of grid-based static scene representation.
The existing anti-alias methods like Mip-NeRF still require intensive computations, while current grid/factorization-based methods don't show good anti-alias results.
This paper proposes a general idea to solve the anti-alias problem: it uses a single-scale grid as the representation while applying learnable convolution to generate multi-scale grids.
Combining with scale-aware position encoding, this paper shows superior results compared to TensoRF or KPlanes, demonstrating a universal improvement for grid-based representation.

**Strengths:**

1. This paper studies an interesting problem in an important direction. Extending the anti-alias idea from Mip-NeRF to explicit representation could contribute to the whole community, and the results are promising. I think it would be beneficial to the development of explicit representations.
2. The whole method is neat, simple, and powerful. Applying CNN on the single grid and having a scale-related positional encoding is simple and intuitive. The overall framework is neat without too many tricks and complicated tweaks. Considering it, this idea could be broadly used in many situations.
3. The paper is overall well-written and easy to follow.

**Weaknesses:**

1. It would be more convincing to compare several baselines with multi-resolution/multi-scale designs. For example, Instant-NGP[17] uses a multi-resolution hash. Also, the original K-Plane uses multi-resolution planes. It would be helpful to show how these method works and how the proposed method compared to these baselines.
2. I am not fully convinced by the analysis of multi-scale and multi-resolution in Section 3.2. To me, the multi-scale one sounds like optimization multi-NeRF at different scales, while NeRFs from different scales are not related. Could you elaborate a little bit? It would be nice to show the experimental results of multi-resolution ideas.
3. Not necessarily a weakness. It would be good to write the method names instead of baseline 1 and 2 in Figure 3.
4. In Table 1, Mip-Kplanes's PSNR has different trends to MIP-NeRF's PSNR and mip-TensoRF's PSNR, also different from Mip-Kplanes' SSIM and LPIPS. Could you elaborate on why  PSNRs are lower with lower Res?

**Questions:**

As stated above, it would be good to show (1) more baselines, especially baselines using multi-scale/resolution methods. (2) experiments compared to multi-resolution grids.

**Limitations:**

I think the limitations discussed in this paper make sense.
I agree the current method is still preliminary and having MIP-NeRF style design would be an exciting direction.

---

> ### Author Rebuttal · Authors · 2023-08-10
>
> **Clarification on multi-resolution, multi-scale grids, and multi-resolution images**
> Multi-resolution grids refer to a set of grids, each of which has a different grid resolution; instant-NGP and K-planes are two examples.
> The opposite notion would be a single-resolution grid, such as the original TensoRF.
> To tackle the issue of mip-mapping, we have proposed a method to generate multi-scale grids by convolving shared representations with learnable kernels, which can be applied to both single- and multi-resolution grids.
> TensoRF is an example of single-resolution-single-scale grids, and K-planes is an example of multi-resolution-single-scale grids, which has already been reported in the main text.
> Similarly, mip-TensoRF is an example of single-resolution-multi-scale model, but the multi-scale grids are generated from the shared representations, which is different from a naive single-resolution-multi-scale model that has multiple explicit grids.
> What we wanted to show was that our method can be directly applied to various grid-based models without modifying other components, regardless of whether they are using single- or multi-resolution grids.
>
> As you suggested, we trained a variant of the single-resolution-multi-scale model, dubbed multi-scale TensoRF, and reported the evaluation results in Table 1.
> Although multi-scale TensoRF outperforms mip-TensoRF at image resolutions of 800 and 400, it requires significantly more extra parameters (4 times that of TensoRF).
> One interesting thing is that mip-TensoRF significantly outperforms multi-scale TensoRF at resolutions of 200 and 100, suggesting shared parameters are not only efficient but also give better representations for rendering low-resolution images.
> Due to limited resources, we were not able to include multi-resolution, multi-scale models (e.g., multiple K-planes) in the rebuttal, but we will certainly add them in the final version.
>
> **Different trends of mip-K-planes and mip-NeRF**
>
> The differential trends between the two approaches can be traced to different experimental details. There are a number of possible causes for this phenomenon:
> To begin, TensoRF employs single-resolution grids, while K-Planes uses multi-resolution grids.
> There are also notable distinctions in the regularization losses implemented by TensoRF and K-Planes.
> Moreover, while K-Planes integrate mixed precision during training, TensoRF relies exclusively on 32-bit precision.
> We had to apply kernel normalization to mitigate the occurrence of explosive gradients and parameters in K-Planes.
> We believe this factor plays a pivotal role in contributing to the observed dissimilarity in trends.

---

> > ### Comment · Reviewer_QWBQ · 2023-08-12
> > **Reply to Rebuttal**
> >
> > Thanks for the reply and it indeed helps.

---

### Official Review · Reviewer_8s5r · 2023-07-05

**Soundness:** 3 good
**Presentation:** 3 good
**Contribution:** 3 good
**Rating:** 5
**Confidence:** 4

**Summary:**

This paper proposes mip-Grid to integrate anti-aliasing techniques into grid-based representations for radiance fields. Mip-Grid generates multiple grids by applying convolution operations over the shared single-scale grid representation. Then it uses the scale-aware coordinate to retrieve the appropriate features from the generated multiple grids. This anti-aliased grid representation combines grid-based representations for faster training and Mip-NeRF-like multi-scale representations for anti-aliasing. The key idea is to use a shared single-scale grid representation and a single sampling but generate multiple grid representations by applying convolutions to the shared single-scale grid representation. Then, the features are extracted by interpolating the generated multiple grid representations using an additional input coordinate.

**Strengths:**

This work mitigates the aliasing artifacts when rendering images from varying camera distances, such as blurry and jaggies in the images, while enjoying fast training time.

This work uses a single-scale shared grid representation and a single-sampling approach, which only introduces minimal additions to the model parameters and computational costs.

The convolutions to generate multiple grids are lightweight and do not incur substantial computational and memory costs as it operates on reduced dimensions.

This work shows comparable performance to mip-NeRF on multi-scale datasets while achieving significantly faster training time.

This work improves the rendering performance of TensoRF and K-Planes on multi-scale datasets while introducing tolerable training time.

**Weaknesses:**

Given the fact that Zip-NeRF is implemented on the hash-based Instant-NGP, thus it achieves fast training time despite using multisampling techniques. I will concern about the contribution of this work, as Zip-NeRF shows good performance for both anti-aliasing and training time. I would like to see the attitude of other reviewers.

I believe "unbounded" scenes are the current research topics for Anti-Aliased NeRFs. Though I understand that addressing these scenarios would require other techniques, the lack of experiments of unbounded scenes (such as Mip-NeRF 360) will degree the rating of this work.

Lack of visualization videos.

**Questions:**

I would suggest including the comparison with Zip-NeRF since an unofficial PyTorch implementation of Zip-NeRF is available.

**Limitations:**

Lack of experiments of unbounded scenes (such as Mip-NeRF 360).

---

> ### Author Rebuttal · Authors · 2023-08-10
>
> **Comparison against Zip-NeRF**
> Zip-NeRF [2] achieves great visual quality at different image resolutions while decreasing the training time to under an hour for the multi-scale 360 dataset [1].
> However, we want to remind reviewers that the authors of Zip-NeRF used 8 V100 GPUs to train their model.
> In our experiments, it took more than 5 hours to train Zip-NeRF on each scene of the multi-scale Blender dataset with a single A100 GPU.
> Ours take less than an hour on the same GPU, achieving better performance in PSNR (Table 1).
> Please note that we used an unofficial code for the experiments, and, thus, these should not be regarded as conclusive results.
> Moreover, our implementation has not been fully optimized yet, and we strongly believe there is substantial room for improvement in training speed, e.g., by using fused-cuda kernels.
>
> **Lack of experiments on unbounded scenes**
> We agree that unbounded scenes are the main focus of current research on anti-aliased NeRF.
> Although it does not cover the scene in 360 degrees, the LLFF dataset consists of real-world images, which are also unbounded scenes.
> Table 3 shows that mip-TensoRF (disc.) significantly outperforms baseline models in all three metrics, including mip-NeRF, on the LLFF dataset.
> Moreover, we trained K-planes and mip-K-planes on the garden scene of the multi-scale 360 dataset [1].
> We used the same contraction function as adopted in K-planes and did not change any hyper-parameters of K-planes regarding both training and network configurations on the LLFF dataset.
> Early results show that our method improves the PSNR of the baseline model at two lower resolutions (Table 6).
> Although our primary model is still behind other models targeting unbounded 360 scenes, there are many fancy techniques that are widely used to extend NeRF-like models to unbounded scenes.
> We strongly believe our method can be greatly improved by adopting those techniques.
> However, building such a model for unbounded scenes requires significant modifications, which pose a challenge that extends beyond the scope of our current paper.
> Considering the current scope and focus of our research, we leave the exploration of this particular aspect for future investigations.
>
>
> **Reference**
>
> [1] Johnatan T. Barron, et al. Mip-nerf 360: Unbounded anti-aliased neural radiance fields. CVPR, 2022.
>
> [2] Johnatan T. Barron, et al. Zip-nerf: Anti-aliased grid-based neural radiance fields. arXiv, 2023.
>
> [3] Sara Fridovich-Keil, et al. K-planes: Explicit radiance fields in space, time, and appearance, CVPR, 2023.

---

> > ### Comment · Reviewer_8s5r · 2023-08-19
> > **Reply**
> >
> > I maintain my rating. Thanks to the authors for the extra details in the rebuttal, particularly the comparison between Zip-NeRF in Table 1 and the multi-scale 360 evaluation in Table 6. It would be helpful to include these in the camera-ready paper/supplement.

---

> ### Author Response · Authors · 2023-08-19
> **Comment by Authors**
>
> We recently noticed that Table 6 was mistakenly omitted from the pdf file. We do apologize for not providing complete materials within the rebuttal period. We include Table 6 at the end of this comment to help reviewers understand our response, especially the second paragraph of our response to reviewer 8s5r. Our model improves the average PSNR of the baseline model by 0.8 in the garden scene of the multi-scale 360 dataset. We used the same contraction function as adopted in K-Planes, and the model configuration and training hyper-parameters remained unchanged. Please note that this is a very early result, and we believe the performance can be greatly improved by adopting many techniques that are widely used to extend NeRF-like models to unbounded 360 scenes. We hope this will clarify any confusion the reviewers may have while reading our response. Please let us know if anything is unclear or if you have any questions.
>
> ---
> Table 6: Evaluation results on the garden scene of the multi-scale 360 dataset.
> | Model| Avg. | Full Res. | 1/2 Res. | 1/4 Res. | 1/8 Res. |
> |:------:|:------:|:------:|:------:|:------:|:------:|
> |K-Planes|24.05|**23.32**|**25.02**|25.45|22.42|
> |mip-K-Planes|**24.86**|22.23|24.26|**25.91**|**27.04**|

---

### Official Review · Reviewer_9wTm · 2023-07-06

**Soundness:** 3 good
**Presentation:** 3 good
**Contribution:** 3 good
**Rating:** 6
**Confidence:** 5

**Summary:**

This work proposes a simple anti-aliasing method for NeRF representations based on factorized grids, e.g., TensoRF and K-planes. The key idea is to learn a lowpass filtering kernel

**Strengths:**

1. The proposed method seems simple and easy to reproduce. Example implementation is also included in the supplementary.
2. The method is evaluated against baseline methods without the proposed anti-aliasing technique, and clear improvements in anti-aliasing can be seen both quantitatively and qualitatively.

**Weaknesses:**

1. It doesn't seem clear how the multi-scale grid is used exactly in rendering; for example, are all the scales used to get a point's feature, or only a heuristically selected scale is used?

2. One thing I'm a bit unsure is that: why does the same kernel size seem to be used across all scales? Would it makes more sense to use bigger kernel size for lower-resolution grids?

3. A visualization of the kernel weights will be helpful in understanding what filtering patterns have been learned, especially the relationship of the grid scale and learned kernel weights.

4. In table 1 and table 2, best models should be highlighted to facilitate reading.


**Questions:**

See the first and second bullet points in the weakness section.

**Limitations:**

The proposed method doesn't seem to be easily applicable to the multi-resolution hash grid proposed by Instant-NGP. It might be better to mention this.

---

> ### Author Rebuttal · Authors · 2023-08-10
>
> **Explanation of how scale coordinates are used**
> Please refer to the global response.
>
> **Same size of kernels across all resolutions**
> We deliberately use the same size of kernels to boost training and inference speeds by leveraging group convolution.
> However, it makes sense to use small kernels for high-scale features and large kernels for low-scale features.
> We trained mip-TensoRF (disc.) on a multi-scale Blender dataset with 3 sets of different kernel sizes: [5, 7, 9, 11], [11, 13, 15, 17], and [3, 5, 9, 17].
> We found the difference in performance to be insignificant, and the kernel sizes barely affected the final output (Table 5).
> This suggests that, even if kernel sizes are the same, kernels for low-scale features can be optimized to have large standard deviations, and vice versa.
>
> **Kernel visualization**
> We visualize the appearance grid kernels of mip-TensoRF (disc.) trained on the Lego scene of the multi-scale Blender dataset to show how the learned patterns of different kernels vary with scale.
> We select the first 16 channels from each resolution and bilinear upsample the kernels by a factor of 8.
> Although we do not provide visualization of density kernels due to the limited space, the learned patterns are very similar to those of appearance kernels.
> As can be seen in Figure 1, the kernel of the highest-scale features looks similar to a Gaussian distribution, and its shape (i.e., standard deviation) becomes larger as the corresponding feature scale decreases.
> This is consistent with our intuition that the kernel of the low-scale features should be smoother than that of the high-scale features and shows that our method learns optimal Gaussian kernels for each scale.
>
> **How to apply the proposed method to Instant-NGP**
> The main focus of our work is to make multiple sets of grid features that can be rendered into different resolutions of images without aliasing artifacts while minimizing the number of additional parameters.
> Filtering the shared representations with learnable kernels is one of many options to achieve this goal.
> In this sense, we can achieve the same goal with Instant-NGP by adding extra hash tables only at the top level and sharing the remaining levels.
> Each table at the top level will learn features suitable for a given scale, and the remaining levels will learn information shared across different scales.
> This approach will also require significantly fewer extra parameters than using multiple hash tables at all levels.
> Although we believe this is an exciting research direction, we leave it as future work as it is hard to build such a model in a short time with limited resources.
>
> **Minor issues**
> We will highlight the best models in all tables.

---

### Official Review · Reviewer_rA7o · 2023-07-07

**Soundness:** 2 fair
**Presentation:** 2 fair
**Contribution:** 2 fair
**Rating:** 6
**Confidence:** 3

**Summary:**

The paper proposes a method to deal with the aliasing issue in factorised grid-based NeRF methods, e.g. TensoRF and K-plane, by applying convolution operations to decomposed vectors and matrices.

---
**After rebuttal**: I have read authors' rebuttal and it addresses my concerns.

**Strengths:**

* The goal of handling aliasing in grid-based NeRF is valuable and the proposed method is novel.
* Source code is provided in supp mat.

**Weaknesses:**

About evaluation:
* I am interested to see what is the performance difference between using convolution operations and using a simple Gaussian?

About methodology:
* I got the impression that the proposed method can only handle a number of *discrete* scales after training, since the convolution kernels are optimised for each scale. If I understand correctly, that means a set of convolution kernels is optimised for a specific scale. If a NeRF is optimised for $4$ scales, for example, scale=$1, \frac{1}{2}, \frac{1}{4}$, and $\frac{1}{8}$, then during inference, it’s impossible to render an image for scale=$\frac{1}{3}$ and $\frac{1}{6}$ from this NeRF?
* Following the point above, if my understanding above is correct, it would be much better to be able to render images from continuous scales.

**Questions:**

See the weakness section above.

**Limitations:**

Yes.

---

> ### Author Rebuttal · Authors · 2023-08-10
>
> **Comparison between learnable and fixed Guassian kernel**
> We trained mip-TensoRF (disc.) with fixed Gaussian kernels on multi-scale Blender dataset and compared them against the same model with learnable kernels.
> We initialized both models with 2 sets of standard deviations [1.0, 1.0, 1.0, 1.0] and [1.0, 1.5, 2.5, 4.0], where the second one aligns with our intuition that low-scale features should be smoothed more than high-scale features.
> Given the fact that the Gaussian kernels initialized with the second set significantly outperform the other (Table 4), the increasing standard deviations appear to be a good inductive bias, but they may not be the optimal kernels for each scale.
> Instead of using heuristically selected kernels, our method fine-tunes the Gaussian kernels and finds the optimal weights during the training.
> The experimental results also show the learnable approach is better than the fixed Gaussian (Table 4).
>
> **Model evaluations at continuous resolutions**
> As briefly mentioned in the global response section, our model can render images at arbitrary resolutions even when trained with the discrete scale coordinates.
> Although the discrete scale coordinates force the model to make features that corresponds to a small number of given scales, the different scales of features can be smoothly interpolated as they share many similarities.
> The problem occurs when the image resolution is reduced below 400 and the difference between features increases accordingly (Table 2).
> However, this minor drawback can be mitigated by using the continuous scale coordinates and additional distance coordinates.
> Table 2 shows that our two revised models can render images at a wide range of resolutions without degrading the visual quality.

---

> > ### Comment · Reviewer_rA7o · 2023-08-16
> > **Quick clarification about 'Continuous scale coordinates'**
> >
> > Thanks for the response. Just double checking: is the **continuous scale coordinates** $s'$ introduced in the rebuttal stage, after submission?

---

> ### Author Response · Authors · 2023-08-17
> **Comment by Authors**
>
> Thanks for the reponse, and indeed, you are correct. The continuous scale coordinate is introduced in the rebuttal stage. However, we want to emphasize that the model trained with the discrete scale coordinates can also render images at arbitrary resolution (Table 2). The reason we provide additional models is to give reviewers an idea of how different kinds of scale coordinates can be applied to our method. Please let us know if you have any other concerns.

---

> > ### Comment · Reviewer_rA7o · 2023-08-17
> > **Thanks for the clarification**
> >
> > Thanks for the clarification. I don't have any other questions. Will raise my ratings at the final review. Best, reviewer.

---

> > > ### Author Response · Authors · 2023-08-19
> > > **Thanks for appreciating our work**
> > >
> > > Thanks for appreciating our work! By the way, it seems the score has not been updated yet, just in case you forget to update the score, we kindly remind you that the discussion period will be closed on Aug 21st, 1pm EDT. Thanks again, and if you have any further questions or clarifications, we will try our best to answer all inquiries promptly.

---

### Author Rebuttal · Authors · 2023-08-10

We briefly explain how the scale coordinate is defined and introduce two additional scale coordinates to show how different scale coordinates can be applied to our method.

**Continuous scale coordinates**
The models reported in the main text use as the scale coordinate $s$ the mean of the pixel width and height in the world coordinates scaled by $2/\sqrt{12}$ [1].
Since this is a discrete value and unique for each image resolution, it may not handle all scale ambiguities, potentially making it difficult for models to render images at unseen resolution.
We introduce the continuous scale coordinate $s'$, the discrete scale coordinate scaled by the distance from a ray origin to a sampled point on a ray, to improve our method further.
Each sample point has a different scale coordinate, and the per-point scales force the kernels to output features that can be smoothly interpolated to unseen scale.

**2-dimensional scale coordinates**
Although the continuous scale coordinate can effectively guide the model to reason grid features of unseen scale, the scale ambiguity still exists.
Consider two different sample points $p_1$ and $p_2$, distanced from the ray origin by $d_1$ and $d_2$, respectively.
If each point's discrete scale coordinates are $d_2$ and $d_1$, they will have the same continuous scale coordinates $d_1 \times d_2$.
To further reduce the scale ambiguity, we use the distance from the ray origin to the sample point as the second scale coordinate.
We convolve the shared representation with 2 sets of 4 kernels, interpolate each set of features separately according to each scale coordinate, and average the feature sets to get the final representations.
The interpolated representations are decoded into densities and RGB values (using a shallow MLP) and rendered as an image.

**Additional experiments**
We compare three variants of mip-TensoRF against baseline models on the test set of multi-scale Blender dataset [1].
TensoRF and single-scale TensoRF are the original TensoRF [4] trained on single-scale and multi-scale dataset, respectively.
Multi-scale TensoRF is a set of 4 TensoRFs, each of which learns different scales of features.
The three mip-TensoRF models use different kinds of scale coordinates: discrete scales, continuous scales, and distance coordinates (abbreviated in parentheses).
Please note that single-scale TensoRF and mip-TensoRF (disc.) are identical to TensoRF (MS) and mip-TensoRF in the main text, respectively.

Table 1 shows the evaluation results at 4 different resolutions averaged across 8 scenes.
The two mip-TensoRF models trained with discrete and continuous scale coordinates, respectively, show only a marginal difference, when evaluated at training resolutions.
A similar trend is observed in evaluations at unseen resolutions, but there is a clear difference when the image resolution is reduced from 400 to 300 (Table 2), suggesting that the continuous scale coordinates make the model generalize better.
Since our method makes multiple sets of feature grids from a shared representation, the feature sets share many similarities, which allows different features to be smoothly interpolated and rendered into visually pleasing images.
However, more information needs to be removed as the output images get smaller, resulting in relatively large differences between low-scale features.
This explains why there is a slight drop in PSNR at lower resolutions.
Moreover, the model trained with 2-dimensional scale coordinates outperforms mip-NeRF [1] by a clear margin in PSNR and SSIM.

**References**

[1] Jonathan T. Barron, et al. Mip-nerf: A multiscale representation for anti-aliasing neural radiance fields. ICCV, 2021.

[2] Anpei Chen, et al. Tensorf: Tensorial radiance fields. ECCV, 2022.

---

### Decision · Program_Chairs · 2023-09-21

**Decision:**

Accept (poster)

**Comment:**

The paper presents a method for anti-aliasing neural radiance fields using a grid-based neural representation. Reviewers appreciated the timeliness and simplicity of the method, the quality of the results, and the clarity of the writing. Some concerns were raised about how the method compares to approaches based on Instant-NGP, TensoRF, and K-Planes. The authors provided additional comparisons in the rebuttal, and post-discussion, all reviewers are positive about acceptance. The AC agrees that the paper meets the bar for acceptance. The authors should include the clarifications and additional results, including the discussion of scale coordinates and the multi-scale Blender dataset experiments, in the camera ready version.